# Programmed Death Ligand 1 Expression in Circulating Tumor Cells as a Predictor and Monitor of Response to Atezolizumab plus Bevacizumab Treatment in Patients with Hepatocellular Carcinoma

**DOI:** 10.3390/cancers16091785

**Published:** 2024-05-06

**Authors:** Takuto Nosaka, Yosuke Murata, Yu Akazawa, Tomoko Tanaka, Kazuto Takahashi, Tatsushi Naito, Hidetaka Matsuda, Masahiro Ohtani, Yoshiaki Imamura, Yasunari Nakamoto

**Affiliations:** 1Second Department of Internal Medicine, Faculty of Medical Sciences, University of Fukui, Fukui 910-1193, Japan; nosat@u-fukui.ac.jp (T.N.); yosukem@u-fukui.ac.jp (Y.M.); aka0124@u-fukui.ac.jp (Y.A.); kawakami@u-fukui.ac.jp (T.T.); tkazuto@u-fukui.ac.jp (K.T.); naitot@u-fukui.ac.jp (T.N.); hidem@u-fukui.ac.jp (H.M.); mohtani@u-fukui.ac.jp (M.O.); 2Division of Diagnostic Pathology/Surgical Pathology, University of Fukui Hospital, Fukui 910-1193, Japan; suki@u-fukui.ac.jp

**Keywords:** programmed death ligand 1, circulating tumor cell, hepatocellular carcinoma, immune checkpoint inhibitor, liquid biopsy

## Abstract

**Simple Summary:**

Hepatocellular carcinoma (HCC) represents a significant health challenge, demanding innovative approaches for effective treatment. Despite advances in immune checkpoint inhibitor (ICI) therapy, reliable biomarkers to predict treatment response in HCC have yet to be found. Our study addressed this critical research gap by exploring the role of circulating tumor cells (CTCs) and their PD-L1 RNA expression in predicting response to atezolizumab and bevacizumab (Atezo/Bev) treatment. CTC-derived RNA collected serially during Atezo/Bev treatment suggests that patients with higher levels of PD-L1 expression in CTCs at baseline were more responsive to this treatment, but not lenvatinib treatment. Moreover, PD-L1 RNA levels in CTCs were an accurate response predictor and a monitorable biomarker that changed dynamically to reflect the response during Atezo/Bev treatment. The results of this study highlight the potential for individualized treatment decisions based on PD-L1 expression in CTCs and have implications for clinical practice.

**Abstract:**

There remains no reliable biomarker of therapeutic efficacy in hepatocellular carcinoma (HCC) for the PD-L1 inhibitor atezolizumab and bevacizumab (Atezo/Bev). Circulating tumor cells (CTCs) enable the serial collection of living tumor cells. Pre-treatment and serial CTC gene expression changes and tumor histology were evaluated to identify predictors of response to Atezo/Bev. Peripheral blood from 22 patients with HCC treated with Atezo/Bev and 24 patients treated with lenvatinib was serially collected. The RNA expression in CTCs was analyzed using qRT-PCR. Higher PD-L1 expression in pre-treatment CTCs was associated with response and improved prognosis with Atezo/Bev treatment, but not with lenvatinib. There was no correlation between PD-L1 expression in CTCs and that in liver tumor biopsy specimens scored using imaging software. Furthermore, PD-L1 RNA expression in CTCs was dynamically altered by Atezo/Bev, decreasing during effective response and increasing upon progression. CTC-derived RNA collected during Atezo/Bev indicates that patients with higher PD-L1 expression in CTCs at baseline were 3.9 times more responsive to treatment. Therefore, PD-L1 RNA levels in CTCs are an accurate response predictor and may be a monitorable biomarker that changes dynamically to reflect the response during Atezo/Bev treatment.

## 1. Introduction

Liver cancer is the sixth most common neoplasm and the third leading cause of cancer-related death, with hepatocellular carcinoma (HCC) constituting approximately 85–90% of these cancers [1]. With advancements in systemic therapy for advanced HCC, immune checkpoint inhibitor (ICI)-based treatments have emerged as pivotal components of systemic first-line therapy [2]. The combination of the programmed death ligand 1 (PD-L1) inhibitor, atezolizumab, and the anti-vascular endothelial growth factor (VEGF) antibody, bevacizumab, has shown promising results for unresectable HCC [3,4]. However, reliable biomarkers for predicting treatment efficacy or monitoring disease progression have not yet been identified [5].

PD-L1 is expressed in a variety of cells, including tumor and immune cells. In some cancer types, such as non-small cell lung cancer (NSCLC), PD-L1 expression in tumor tissues is considered an effective biomarker for predicting the efficacy of ICIs [6,7]. However, in the phase 2 CheckMate 040 and KEYNOTE 224 studies, PD-L1 expression in tumor tissue was not a predictor of response to ICI therapy in patients with HCC [8,9]. Stratification of patients with HCC using biomarkers for optimal response to ICI-based therapy remains an unmet imperative.

Blood-based tumor-derived materials offer a real-time, minimally invasive alternative for early detection, prognostication, and prediction of responses to anticancer agents [10]. Circulating tumor cells (CTCs) shed into the bloodstream from primary and/or metastatic tumors enable the serial collection of live tumor cells, contributing to a more profound understanding of tumor biology [11]. Cancer cells can undergo epithelial-to-mesenchymal transition (EMT) to facilitate their detachment from the primary tumor and intravasation into the blood circulation [12,13]. The characteristics of CTCs differ from those of tumor cells in primary tumors because EMT involves the loss of epithelial characteristics, for example, downregulation of the adhesion molecule E-cadherin, and the acquisition of mesenchymal characteristics, including expression of the cytoskeletal protein vimentin [12,13]. We hypothesized that serial profiling of CTCs before and during atezolizumab plus bevacizumab (Atezo/Bev) treatment in patients with HCC may reveal molecules associated with treatment response and resistance.

In this study, to investigate the predictors of response to Atezo/Bev in patients with HCC, we evaluated pre-treatment gene expression in CTCs, tumor histology, and clinical features. Among these factors, higher PD-L1 RNA levels in CTCs were associated with a better response and prognosis for Atezo/Bev, but not lenvatinib. Moreover, PD-L1 RNA expression in CTCs exhibited dynamic alterations during Atezo/Bev treatment, decreasing during the effective response and increasing with tumor progression. PD-L1 RNA levels in CTCs may be an accurate predictor of response and a monitorable biomarker that dynamically changes in patients with HCC during Atezo/Bev treatment, reflecting the treatment response.

## 2. Patients and Methods

### 2.1. Study Protocol and Participants

This prospective study was conducted in accordance with the Declaration of Helsinki and approved by the Research Ethics Committee of University of Fukui (20210168). Written informed consent was obtained from all the participants. From September 2020 to December 2023, peripheral blood samples were collected at baseline, at 1–3 weeks, at the first response evaluation, and at multiple follow-up time points in 22 patients treated with Atezo/Bev and 24 patients treated with lenvatinib for unresectable HCC at the University of Fukui Hospital. The clinical characteristics of the patients are listed in Table 1. Percutaneous tumor biopsies were performed on 11 patients before Atezo/Bev treatment. ALBI (albumin-bilirubin) grade was calculated based on serum albumin and total-bilirubin values using the following ALBI-score formula: (log10 bilirubin (µmol/L) × 0.66) + (albumin (g/L) × −0.085). ALBI grade was then defined based on the score and the following scale: ≤−2.60 = Grade 1; >−2.60 to ≤−1.39 = Grade 2; and >−1.39 = Grade 3 [14]. ALBI grade 2 was further divided into two sub-grades (2a and 2b) using a previously reported cut-off value (ALBI score −2.270), and the four ALBI grades were named as modified ALBI grades [15].

### 2.2. Etiology of Liver Diseases

The etiology of HCC was determined in patients testing positive for anti-hepatitis C virus antibody (HCV Ab) as HCV, and those testing positive for the surface antigen of hepatitis B virus (HBV) (HBsAg) were considered HBV-positive. Patients negative for both anti-HCV antibody and HBsAg were defined as non-B-non-C (NBNC) tumors.

### 2.3. Treatment Protocol

Atezolizumab and bevacizumab were administered according to the IMbrave 150 protocol schedule at the following doses: 1200 mg atezolizumab and 15 mg/kg bevacizumab every three weeks until loss of clinical benefit or unacceptable toxicity [3]. Atezolizumab and bevacizumab were withdrawn or reduced to manage adverse events (AEs). The dose of oral lenvatinib was 8 mg/day for patients who weighed < 60 kg and 12 mg/day for patients who weighted ≥ 60 kg. If a grade 2 AE was deemed unacceptable or a grade 3 AE was observed, a dose reduction or temporary interruption of lenvatinib was administered. The treatment response was evaluated using dynamic computed tomography or gadolinium ethoxybenzyl magnetic resonance imaging (Gd-EOB-MRI) at 8–12 weeks after the first administration. Treatment was discontinued if unacceptable adverse events or progressive disease (PD) were observed. Radiological responses were evaluated using the Response Evaluation Criteria for Solid Tumors (RECIST) criteria ver.1.1. The AEs were evaluated and graded according to the National Cancer Institute Common Terminology Criteria for Adverse Events v5.0 (https://ctep.cancer.gov/protocoldevelopment/electronic_applications/ctc.htm, accessed on 26 March 2024).

### 2.4. Evaluation of the Treatment Response

Overall survival (OS) was defined as the time from the date of the first cycle of treatment until death. Progression-free survival (PFS) was defined as the time from the date of the first cycle of treatment to the date of death, or the date of radiological evidence of tumor progression. 

### 2.5. Enrichment of CTCs and RNA Extraction

Peripheral blood samples (10 mL) were collected in an ethylenediaminetetraacetic acid (EDTA)-K2 anticoagulant tube. Before this collection, the first 5 mL of blood was discarded to prevent epithelial cell contamination. CTCs were enriched by negative selection using the RosetteSep Human CD45 Depletion Cocktail (StemCell Technologies, Vancouver, Canada) according to the manufacturer’s protocol. The samples were then incubated with the Depletion Cocktail at 50 µL/mL for 20 min at room temperature. Thereafter, cellular separation was achieved in SepMate 50 mL (StemCell Technologies) tubes containing 15 mL of Lymphoprep density gradient medium (StemCell Technologies). Subsequently, the samples were centrifuged for 5 min at 1200 rpm. The upper phase was transferred to a 50 mL tube and reconstituted to 50 mL with PBS containing fetal bovine serum (Sigma-Aldrich, Saint Louis, MO, USA). After centrifugation at 300× *g* for 10 min, the samples were lysed with an ammonium chloride solution (StemCell Technologies). The cell pellet was then rinsed twice with 50 mL PBS and 2% FBS. For each patient, the enriched cells were equally divided into two halves: one for flow cytometric analysis, and the other for RNA extraction for qRT-PCR. Toral RNA was extracted using an RNeasy Mini Kit (Qiagen, Hilden, Germany). RNA extraction and flow cytometry were performed within 8 h of collection. 

### 2.6. Immunofluorescent Staining

Cells enriched with RosetteSep were fixed with 4% paraformaldehyde and blocked using a blocking antibody diluent (PerkinElmer/Akoya Biosciences, San Diego, CA, USA). The cells were further incubated with an optimal dilution of APC/Cyanine7 mouse anti-human CD45 (2D1) (BioLegend, San Diego, CA, USA), PE mouse anti-human pan-cytokeratin (C-11) (Cayman Chemical, Ann Arbor, MI, USA), or isotype-matched control mouse IgG (BD Biosciences, San Jose, CA, USA). The antibodies used are listed in Table 2. The nuclei were counterstained with 4’-6-diamidino-2-phenylindole (DAPI) (PerkinElmer/Akoya Biosciences). Immunofluorescence was detected using a fluorescence microscope (BZ-X800; Keyence, Itasca, IL, USA).

### 2.7. Flow-Cytometric Analysis

After separation of blood using Rosettesep, enriched cells were labeled with fluorescent dye-conjugated antibodies, APC/Cyanine7 mouse anti-human CD45 (2D1) (BioLegend, San Diego, CA, USA), PE mouse anti-human pan-cytokeratin (cytokeratin 4, 5, 6, 8, 10, 13, and 18) (C-11) (Cayman Chemical), and BV421 mouse anti-human PD-L1 (29E.2A3) (BD Biosciences). Cell pellets were resuspended in 500 μL PBS and counted through multiparametric flow cytometry using a BD FACSAriaII (BD Biosciences). CD45 negative and Pan-CK positive cells were defined as CTCs. The data were analyzed using FlowJo version 10.9.0 (BD Biosciences). Isotype-matched control IgGs for individual mouse monoclonal antibodies (BD Biosciences) and CompBeads anti-mouse Ig, κ/Negative Control Compensation Particles Set (BD Biosciences) were used for compensation. The antibodies used are listed in Table 2. To detect dead cells, 7-aminoactinomycin D (7-AAD; BD Biosciences) was added to the buffer immediately before flow cytometry. In this study, we considered the 7-AAD(−)CD45(−)PanCK(+) populations detected by flow cytometry as CTCs. 

### 2.8. RNA Extraction and Nested-PCR

Total RNA was reverse-transcribed using the High-Capacity cDNA Reverse Transcription Kit (Applied Biosystems, Foster City, CA, USA) to synthesize first-strand complementary DNA, and qPCR was performed using the TaqMan Gene Expression Assay (Thermo Fisher Scientific, Waltham, MA, USA) and StepOnePlus (Applied Biosystems, USA) following pre-amplification reactions of the cDNA using TaqMan PreAmp Master Mix (Applied Biosystems, USA). The primers and TaqMan probes for these genes were obtained from predesigned assays of Applied Biosystems (Table 3). The first RT-PCR reaction was performed with each target using the TaqManTM PreAmp Master Mix. The samples were incubated at 95 °C for 10 min before being subjected to 14 cycles of denaturation at 95 °C for 15 s and annealing at 60 °C for 4 min. The first reaction was performed using a TaKaRa PCR Thermal Cycler Dice Standard (TaKaRa Bio, Shiga, Japan). Samples were used as templates in a second semi-nested RT-PCR amplification performed using StepOnePlus (Applied Biosystems). Briefly, semi-nested PCR amplification was performed in a 10 µL final reaction with each primer and TaqMan probe used in the first RT-PCR. The reaction mixture was preheated at 95 °C for 10 min, followed by 40 cycles of 95 °C for 15 s and 60 °C for 1 min. The expression of target genes was analyzed using the ΔΔCt comparative threshold method. GAPDH was used as an internal control.

### 2.9. Percutaneous Hepatic Tumor Needle Biopsy

HCC tissue specimens were obtained from the center of the tumor before Atezo/Bev treatment using a percutaneous fine-needle aspiration biopsy (Majima needle, 21G; Top, Tokyo, Japan) under abdominal ultrasound guidance. The tissue specimens were fixed in formalin and embedded in paraffin.

### 2.10. Immunohistochemical Analysis

Human liver tissues were fixed in 10% formaldehyde, embedded in paraffin, and stained with hematoxylin and eosin. Simultaneously, the tissues were stained for IHC analysis. For antigen removal, deparaffinized slides were autoclaved in BondTM Epitope Retrieval Solution 2 (Leica Biosystems, Buffalo Grove, IL, USA) at 121 °C for 20 min. Endogenous peroxidase activity was blocked using 0.3% H_2_O_2_ for 15 min, followed by incubation of the slides using an antibody diluent/block (PerkinElmer Akoya Biosciences, CA, USA). Sections were further incubated with rabbit monoclonal [SP142] antibody against human PD-L1 (Abcam, Cambridge, UK) at a dilution ratio of 1:100. The antibodies used are listed in Table 2. The immunoconjugates were detected using Histofine SimpleStain MAX PO (R) anti-rabbit antibody (Nichirei, Tokyo, Japan) in accordance with the manufacturer’s instructions. Samples were examined using a Mantra microscope (PerkinElmer/Akoya Biosciences), and the images were analyzed using inForm® software v2.6 (PerkinElmer/Akoya Biosciences). PD-L1 positive cells in the tumor foci were measured in three to five randomly chosen visual fields.

### 2.11. Automated Histological Image Analysis

Image analysis was performed using inForm® software v2.6 (Akoya Bioscience). Multiple representative images were selected as batch regions for training. Cell segmentation and phenotyping were performed to assign each cell to a phenotypic category. Histological scores (H-scores) were analyzed with the 4-bin algorithm and calculated based on the PD-L1 staining intensity using the following formula: H-score = (1× % of weakly stained cells) + (2× % of moderately stained cells) + (3× % of strongly stained cells).

### 2.12. Statistical Analyses

Statistical significance was determined using the Mann–Whitney U test, Wilcoxon matched pair rank test, or one-way analysis of variance, followed by the Tukey–Kramer post-hoc test. Cumulative survival was analyzed using the Kaplan–Meier method, and differences were analyzed using the log-rank test. The univariate log-rank test was used to determine predictive factors for PFS. Statistical analyses were performed using the GraphPad Prism software (version 10; GraphPad Software Inc., San Diego, CA, USA). Statistical significance was set at *p* < 0.05.

## 3. Results

### 3.1. Association of Pre-Treatment CTC Characteristics, Tumor Tissue, and Clinical Features with Response to Atezo/Bev

To investigate the predictive factors of Atezo/Bev response in patients with unresectable HCC, we analyzed CTC specimens collected from peripheral blood and tumor biopsy specimens prior to treatment (Figure 1A). CTCs were enriched from peripheral blood using Rossettsep, and subsequently, CTC RNA expression was analyzed by qRT-PCR. CD45(−)PanCK(+) cells were counted as CTCs using flow cytometry (Figure 1B). Microscopic images of CD45(−)PanCK(+) CTCs after enrichment with Rossettsep are illustrated in Figure 1C. In 22 patients with HCC treated with Atezo/Bev, the initial response was PR/SD in 14 patients and PD in 8 patients. The following samples, collected prior to Atezo/Bev treatment, were analyzed for their association with the initial treatment response: (1) expression of 14 genes associated with cancer progression in CTC RNA, (2) number of CTCs, (3) tumor differentiation and PD-L1 expression in liver tumor biopsy tissues, and (4) clinical features (Figure 1D). Analysis of (1), (2), and (4) was performed on all 22 patients. For (3), liver tumor biopsy tissue was obtained and analyzed before treatment in 11 of the 22 patients. In the univariate analysis, PD-L1 RNA expression in CTCs was associated with the PFS, whereas other factors were not (Table 4). These results suggest that PD-L1 RNA expression in the CTCs of patients with HCC may be a predictor of the first treatment response to Atezo/Bev treatment.

### 3.2. Pre-Treatment PD-L1 RNA Level in CTCs and Response to Atezo/Bev and Lenvatinib

We evaluated the association between the first treatment response and RNA expression of 14 genes related to immune checkpoint molecules, stem cell-related molecules, EMT-related molecules, and tumor markers in CTCs prior to Atezo/Bev treatment. PD-L1 mRNA expression in CTCs was significantly higher in the PR/SD group than in the PD group (Figure 2A). No significant differences were observed for the other 13 molecules. Twenty four HCC patients treated with lenvatinib, a tyrosine kinase inhibitor, showed no difference in PD-L1 RNA expression levels in pre-treatment CTCs during the treatment response (Figure 2B). The clinical outcomes of Atezo/Bev-treated patients were evaluated in two groups according to PD-L1 RNA levels in CTCs, and PFS and OS were significantly better in the group with higher PD-L1 expression (Median PFS, CTC PD-L1 High/Low 11.97/3.09 months) (PFS HR 0.33, 95% CI 0.12–0.87, *p* < 0.05; OS HR 0.15, 95% CI 0.038–0.59, *p* < 0.01) (Figure 2C). These results indicate that PD-L1 RNA expression in CTCs prior to Atezo/Bev treatment in patients with HCC may be a valuable biomarker for predicting treatment response and prognosis.

### 3.3. PD-L1 Expression in CTCs and Matched Tumor Tissues

We evaluated the association between PD-L1 protein expression in liver tumor biopsy specimens and PD-L1 RNA expression in CTCs prior to Atezo/Bev treatment. HCC tissue specimens were obtained from the center of the tumor using a percutaneous fine-needle aspiration biopsy under abdominal ultrasound guidance. PD-L1 IHC staining of liver tumor biopsies was performed using the inForm image analysis software to calculate H-scores (Figure 3A). No correlation was found between the PD-L1 H-score of liver tumor biopsies and PD-L1 RNA expression in CTCs (Figure 3B). There was no difference in the PD-L1 H-score of liver tumor biopsies relating to the Atezo/Bev treatment effect (Figure 3C). These results indicate that the intensity of PD-L1 expression in biopsy specimens of primary liver tumors does not correlate with PD-L1 expression in CTCs and is poorly associated with the therapeutic response to Atezo/Bev.

### 3.4. Association between Dynamic Changes in PD-L1 RNA Level in CTCs and Response during Atezo/Bev and Lenvatinib

To monitor the long-term changes in PD-L1 RNA expression in CTCs, CTCs were serially collected during Atezo/Bev treatment. In PR/SD cases, PD-L1 expression in CTCs was lower than that at the initiation of treatment, and PD-L1 levels remained low in many cases with subsequent sustained responses (Figure 4A,C). Among the five patients who subsequently had a PD response, all five patients exhibited elevated PD-L1 levels compared with the previous CTC collection. In patients with PD as the initial response to Atezo/Bev treatment, PD-L1 expression was increased in all eight patients compared to pre-treatment. However, in patients with HCC treated with lenvatinib, there was no consistent change in PD-L1 expression in CTCs depending on the treatment response (Figure 4B,C). These results suggest that, unlike lenvatinib treatment, PD-L1 expression in CTCs is dynamically altered by Atezo/Bev treatment, decreasing the effective response and increasing progression.

### 3.5. Pre-Treatment Levels and Dynamic Changes of PD-L1 RNA in CTCs during Atezo/Bev

The clinical course of a representative case was as follows: The first patient (case #17, 64 y.o., M, 3.0 cm, multinodular, BCLC-B, well differentiated) was classified in the high PD-L1 expression CTC group and low PD-L1 expression liver tumor biopsy group (Figure 5A). With Atezo/Bev treatment, the patient was evaluated for SD after 2.4 months, but showed a PD response at 5.5 months. After lenvatinib treatment, the patient was treated with Atezo/Bev again and showed a PR in mRECIST (SD with decreased early staining) SD response. The PD-L1 expression in CTCs was unchanged in the SD response, but increased at the time of PD and decreased after re-treatment with Atezo/Bev. Flow cytometric analysis of CTCs showed a change in the proportion of PD-L1 (+) CTC, consistent with the changes in PD-L1 RNA expression in CTCs. The second patient (case #2, 64 y.o., M, 5.5 cm, multinodular, BCLC-C, moderately differentiated) was classified in the low PD-L1 expression CTC group and high PD-L1 expression liver tumor biopsy group (Figure 5B). After 2.8 months of Atezo/Bev treatment, the initial response showed PD. PD-L1 expression in CTCs was increased in patients with PD compared to pre-treatment levels. These results suggest that the efficacy of Atezo/Bev treatment is predictable according to the PD-L1 RNA expression level of CTCs before treatment, independent of the PD-L1 level in hepatic tumor tissue. Moreover, the PD-L1 expression of CTCs dynamically changes in response to subsequent treatment effects.

## 4. Discussion

To investigate the predictors of response to Atezo/Bev in patients with HCC, pre-treatment CTC characteristics, tumor histology, and clinical features were evaluated. Among these factors, higher PD-L1 expression in CTCs was associated with a better response and prognosis for Atezo/Bev, but not for lenvatinib. In liver tumor tissue, PD-L1 levels did not correlate with those of CTCs. Furthermore, PD-L1 RNA expression in CTCs was dynamically altered by Atezo/Bev treatment, decreased during the effective response, and increased upon progression.

The PD-L1 expressing CTCs were first evaluated in breast cancer [16] and subsequently in cancers of the lungs [17,18], melanoma [19], head and neck squamous cell carcinoma [20], colon [21], prostate [21], and liver [22]. Meta-analyses have shown that cancer patients with CTCs expressing PD-L1 have a poorer prognosis and PFS [23]. Interestingly, several studies have reported favorable therapeutic effects of ICI therapy in patients with cancer who have CTCs expressing PD-L1 in NSCLC [24,25], urothelial carcinoma [26], melanoma [27], and HCC [22]. Conversely, some reports indicate that it is a marker of low efficacy of ICI therapy [27,28,29]. These differences may be due to variations in carcinomas, methods of collecting CTCs, and methods of assessing PD-L1 expression [23]. We observed for the first time that pre-treatment PD-L1 RNA expression may predict the PFS and OS of Atezo/Bev treatment in patients with HCC.

In this study, PD-L1 expression in CTCs changed dynamically with Atezo/Bev treatment, decreasing during the effective response and increasing in progression. Consistent with this, PD-L1-expressing CTCs have been reported to decrease during ICI treatment in an effective response, increasing upon treatment resistance in NSCLC [18,24,28,30,31], urothelial carcinoma [26], and HCC [29]. Cancer cells expressing PD-L1 bind to the PD-1 receptor on T cells and inhibit their proliferation and cytotoxic activity [32]. Spiliotaki et al. suggested that CTCs expressing PD-L1 suppress T cell immune responses, leading to resistance to ICI therapy [31]. It has also been reported that cancer cells expressing PD-L1 in tumor tissues have more aggressive characteristics [33]. Therefore, cancer cells that migrate into the blood during ICI therapy may express PD-L1 to evade the immune response [18].

No correlation was observed between PD-L1 expression scores of liver tumor tissues and RNA expression in CTCs. Various studies have reported no correlation between PD-L1 expression in tumor tissues and CTCs in NSCLC [28,30,34], urothelial carcinoma [26], and breast cancer [35]. These clinical results support the notion that PD-L1 expression in cancer cells is a heterogeneous and dynamic biomarker that varies spatially and temporally [36,37]. As PD-L1 expression varies between primary and metastatic sites and even within tumor nests, PD-L1 expression analysis of CTCs is considered a useful tool to complement this heterogeneity [34]. We are the first to observe no correlation in PD-L1 expression between primary HCC tissue and CTCs.

The present study had several limitations. First, we analyzed CTCs from 22 patients with HCC treated with Atezo/Bev, which is not sufficient to draw definitive conclusions. However, we evaluated the significance of PD-L1 expression analysis of CTCs in Atezo/Bev treatment by comparing 24 patients with HCC treated with lenvatinib and serially collected CTCs during treatment. Second, this study focused on stem cells, EMT, and tumor markers of CTC gene expression. In future studies, we plan to perform single-cell whole-genome analysis for further examination.

## 5. Conclusions

Analysis of CTC-derived RNA collected serially during Atezo/Bev treatment in patients with HCC indicated that patients with higher PD-L1 expression in CTCs at baseline were 3.9 times more responsive to treatment. PD-L1 RNA levels in CTCs may be an accurate predictor of response and a monitorable biomarker that dynamically changes in patients with HCC during Atezo/Bev treatment, reflecting the response.

## Figures and Tables

**Figure 1 cancers-16-01785-f001:**
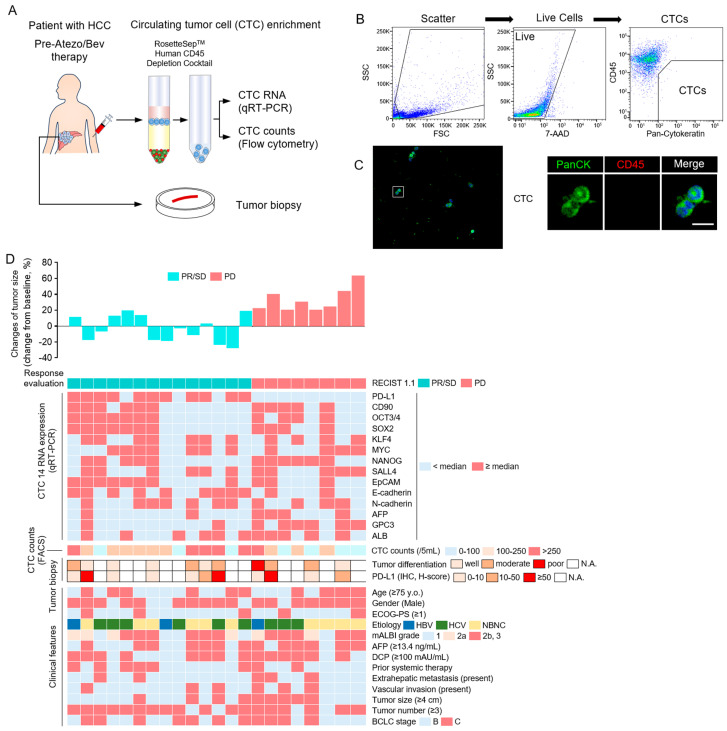
Association of pre-treatment CTC characteristics, tumor histology, and clinical features with response of HCC patients to Atezo/Bev. (**A**) Peripheral blood of patients was collected before Atezo/Bev treatment, and CTCs were enriched using Rosettesep™. The CTCs gene expression was analyzed using qRT-PCR, and expression of surface protein was analyzed via flow cytometry. Liver tumor biopsies were performed before Atezo/Bev treatment. (**B**) Flowchart showing CTC isolation of HCC patients by multiparametric flow cytometry. CTCs were analyzed by flow cytometry in two steps: (i) removal of dead cells by 7-AAD, and (ii) isolation of PanCK(+) CD45(−) cells as CTCs. (**C**) Immunofluorescence staining of CTC enriched by Rosettesep™. PanCK, CD45, and DAPI were stained. A representative image is presented. Bar, 10 μm. (**D**) Heatmap of pre-treatment factors and initial treatment response and changes of tumor size (%) in 22 HCC patients treated with Atezo/Bev: (1) 14RNA expression of CTCs measured by qRT-PCR, (2) number of CTCs measured by FACS, (3) histological evaluation of liver tumor biopsy, and (4) clinical characteristics.

**Figure 2 cancers-16-01785-f002:**
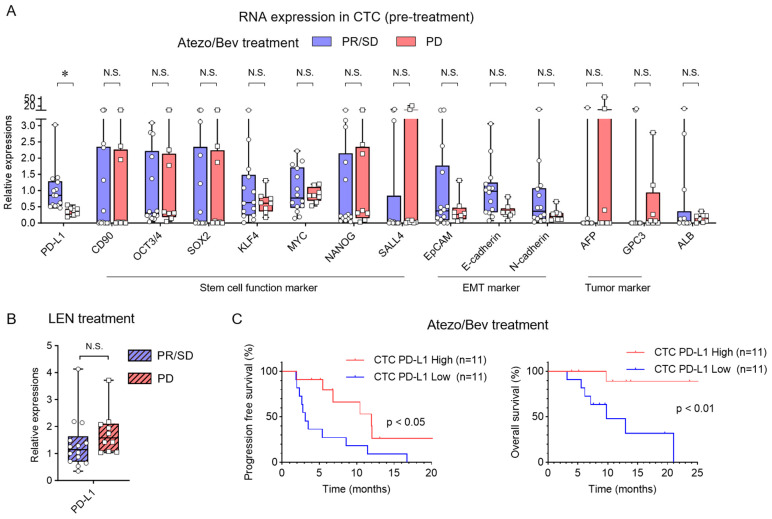
Pre-treatment PD-L1 RNA levels and response to Atezo/Bev and lenvatinib in CTCs. (**A**) The expression of 14 genes in the pre-treatment CTC mRNA of HCC patients treated with Atezo/Bev [PR/SD (n = 14) and PD (n = 8)] was analyzed by real-time qRT-PCR. (**B**) The PD-L1 expression in pre-treatment CTC mRNA of HCC patients treated with lenvatinib [PR/SD (n = 14) and PD (n = 10)] was analyzed via real-time qRT-PCR. (**C**) HCC patients were divided into High (n = 11) and Low (n = 11) expression groups by PD-L1 expression level in pre-treatment CTC. Kaplan–Meier curves of progression-free survival and overall survival for patients in the High and Low PD-L1 expression groups in pre-treatment CTC. (**A**,**B**) Mann–Whitney test. (**C**) Log-rank test. * *p* < 0.05, N.S., not significant.

**Figure 3 cancers-16-01785-f003:**
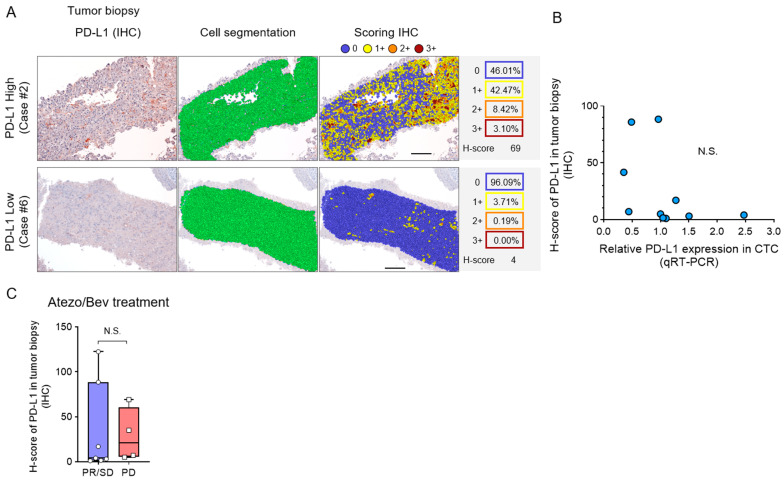
PD-L1 expression in CTCs and matched liver tumor tissues. (**A**) Representative PD-L1 stained image of liver tumor biopsy tissue. The image was captured with a Mantra microscope, and the PD-L1 histological score (H-score) was calculated with inForm software. The left panel presents the PD-L1 microscopic image, cell segmentation image, scoring image, and H-scores analyzed via the 4-bin algorithm according to staining intensity. Tissue images of representative cases with high H-score (top panel) and low H-score (bottom panel) are illustrated. Bar, 100 μm. IHC staining observed and measured at ×200 magnification. (**B**) Correlation between pre-treatment H-score of PD-L1 expression in liver tumor biopsy tissue and PD-L1 expression RNA levels in CTC from HCC patients (n = 11). (**C**) The H-score of PD-L1 in tumor biopsy of HCC patients treated with Atezo/Bev [PR/SD (n = 7) and PD (n = 4)]. (**B**) Simple linear regression analysis. (**C**) Mann-Whitney test. N.S., not significant.

**Figure 4 cancers-16-01785-f004:**
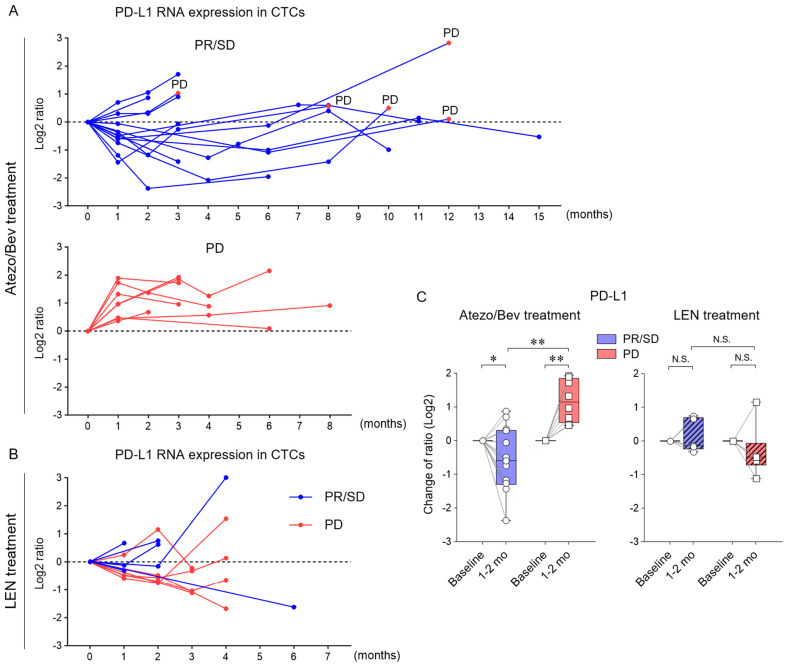
Association of dynamic changes in PD-L1 RNA levels in CTCs with responses during treatment with Atezo/Bev and lenvatinib. (**A**) Changes in PD-L1 RNA expression levels in CTCs in HCC patients treated with Atezo/Bev [PR/SD (n = 14) and PD (n = 8)]. PD-L1 RNA expression levels of CTCs over the treatment course were calculated by dividing the level prior to the treatment in each case and expressed as a log2 ratio. In PR/SD cases, the values at the time of PD are represented by red dots. (**B**) Changes in PD-L1 RNA expression levels in CTCs in HCC patients treated with lenvatinib [PR/SD (n = 6) and PD (n = 6)]. (**C**) Change in PD-L1 RNA expression in CTCs of HCC patients treated with Atezo/Bev [PR/SD (n = 14) and PD (n = 8)] (left) and lenvatinib [PR/SD (n = 5) and PD (n = 6)] (right) from baseline to 1–2 months after treatment. (**C**) Mann–Whitney test. * *p* < 0.05, ** *p* < 0.01, N.S., not significant.

**Figure 5 cancers-16-01785-f005:**
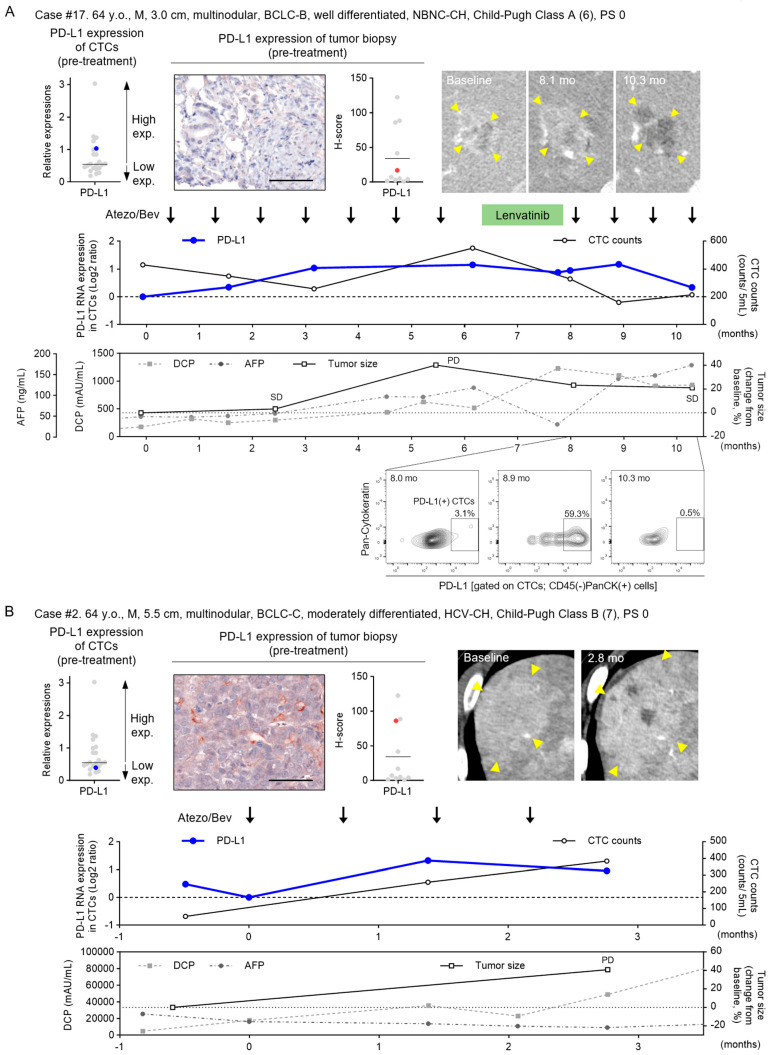
Treatment course of representative HCC patients with pre-treatment levels and dynamic changes of PD-L1 RNA in CTCs during Atezo/Bev treatment. Clinical course of two representative cases treated with atezolizumab and bevacizumab (Atezo/Bev) for HCC. Case #17 was SD at the initial treatment response and subsequently PD but showed SD after rechallenge (**A**), and case #2 showed progressive disease (PD) (**B**). PD-L1 expression level of CTCs before treatment (blue dots), PD-L1 IHC image of liver tumor biopsy, H-score (red dots), imaging course (arterial phase of dynamic CT, yellow arrowheads indicate tumor), change in PD-L1 expression of CTCs, total number of CTCs, tumor diameter, tumor marker (AFP, DCP) values are shown. Gray dots in PD-L1 expression of CTCs and tumor H-score indicate 21 other patients and 10 other patients with HCC treated with Atezo/Bev assayed in this study. The gating images and percentages of PD-L1(+) CTCs (CD45(−) PanCK(+) cells) by FACS analysis are shown in (**A**).

**Table 1 cancers-16-01785-t001:** Characteristics of patients with hepatocellular carcinoma treated with atezolizumab plus bevacizumab, and lenvatinib.

Characteristics	Atezolizumab + Bevacizumab (n = 22)	Lenvatinib (n = 24)
Age, median (IQR), years	73 (64–80)	73 (66–78)
Gender, male/female, n	17/5	19/5
ECOG PS, 0/1/2/3/4, n	17/5/0/0/0	22/2/0/0/0
Etiology, HBV/HCV/NBNC, n	3/9/10	3/10/11
PLT, ×10^9^/L, median (IQR)	151 (110–176)	144 (116–168)
PT, INR, median (IQR)	1.06 (0.98–1.15)	1.03 (0.96–1.10)
ALB, g/dL, median (IQR)	3.5 (3.1–3.8)	3.6 (3.4–4.0)
T-bil, g/dL, median (IQR)	0.9 (0.6–1.3)	0.8 (0.6–1.0)
modified ALBI grade, 1/2a/2b/3, n	5/6/10/1	9/4/10/1
ALT, IU/L, median (IQR)	25 (19–37)	29 (14–40)
AFP, ng/mL, median (IQR)	16.5 (6.0–312.7)	10.8 (3.9–135.6)
DCP, mAU/mL, median (IQR)	602 (26–1569)	275 (30–744)
Maximum tumor size, cm, median (IQR)	3.5 (2.6–6.5)	3.4 (2.2–7.7)
Number of tumor, 1/2/3+, n	2/2/18	1/3/20
Vascular invasion, absent/present, n	17/5	18/6
BCLC stage, A/B/C, n	0/11/11	1/14/9
Extrahepatic metastasis, n		
None	17	20
Lymph node	3	3
Bone	1	0
Lung	1	1
Prior systemic therapy, n		
None	13	12
Sorafenib	1	0
Lenvatinib	5	―
Atezolizumab plus Bevacizumab	―	8
HAIC	1	3
Lenvatinib, HAIC	2	0
Sorafenib, Atezolizumab plus Bevacizumab	―	1
CTC counts (/5 mL), median (IQR)	161 (97–254)	213 (38–422)
Tumor biopsy		
Tumor differentiation, well/moderate/poor, n	5/5/1	―
PD-L1 (IHC, H-score), median (IQR)	7 (4–52)	―
Observation period, median, days	314	261

Abbreviations: AFP, alpha-fetoprotein; BCLC, Barcelona Clinic Liver Cancer; CTC, circulating tumor cell; DCP, des-gamma-carboxy prothrombin; ECOG PS, Eastern Cooperative Oncology Group performance status; HAIC, hepatic arterial infusion chemotherapy; H-score, histological score; IHC, immunohistochemistry; IQR, interquartile range; NBNC, nonB-nonC; PD-L1, programmed death ligand 1.

**Table 2 cancers-16-01785-t002:** Antibodies were used in this study.

Target	Application	Target Species	Host Species	Clone	Company	Catalogue No.
CD45	Flow cytometry; IF	Human	Mouse	2D1	BioLegend	368516
pan-Cytokeratin	Flow cytometry; IF	Human	Mouse	C-11	Cayman Chemical	10478
PD-L1	Flow cytometry	Human	Mouse	29E.2A3	BD Biosciences	568319
PD-L1	IHC	Human	Rabbit	28-8	abcam	ab205921

Abbreviations: IF, immunofluorescence; IHC, immunohistochemistry.

**Table 3 cancers-16-01785-t003:** Primers were used in this study.

Gene	Species	Dye	Company	Catalogue No.
AFP	Human	FAM	Applied Biosystems	Hs00173490_m1
ALB	Human	FAM	Applied Biosystems	Hs00609411_m1
CD274 (PD-L1)	Human	FAM	Applied Biosystems	Hs00204257_m1
CDH1 (E-cadherin)	Human	FAM	Applied Biosystems	Hs01023895_m1
CDH2 (N-cadherin)	Human	FAM	Applied Biosystems	Hs00983056_m1
EpCAM	Human	FAM	Applied Biosystems	Hs00901885_m1
GAPDH	Human	FAM	Applied Biosystems	Hs02786624_g1
GPC3	Human	FAM	Applied Biosystems	Hs01018936_m1
KLF4	Human	FAM	Applied Biosystems	Hs00358836_m1
MYC	Human	FAM	Applied Biosystems	Hs00153408_m1
NANOG	Human	FAM	Applied Biosystems	Hs02387400_g1
POU5F1 (OCT3/4)	Human	FAM	Applied Biosystems	Hs04260367_gH
SALL4	Human	FAM	Applied Biosystems	Hs00360675_m1
SOX2	Human	FAM	Applied Biosystems	Hs01053049_s1
THY1 (CD90)	Human	FAM	Applied Biosystems	Hs06633377_s1

**Table 4 cancers-16-01785-t004:** Univariate analysis of factors associated with progression-free survival of patients with hepatocellular carcinoma treated with atezolizumab plus bevacizumab.

Factors	Patients Number (n = 22)	Univariate Analysis (*p*-Value)
**CTC RNA expression (qRT-PCR)**		
PD-L1, (<median/≥median), n	11/11	**0.018**
CD90, (<median/≥median), n	11/11	0.910
OCT3/4, (<median/≥median), n	11/11	0.929
SOX2, (<median/≥median), n	11/11	0.953
KLF4, (<median/≥median), n	11/11	0.209
MYC, (<median/≥median), n	11/11	0.436
NANOG, (<median/≥median), n	11/11	0.943
SALL4, (<median/≥median), n	11/11	0.576
EpCAM, (<median/≥median), n	11/11	0.502
E-cadherin, (<median/≥median), n	11/11	0.730
N-cadherin, (<median/≥median), n	11/11	0.167
AFP, (<median/≥median), n	16/6	0.683
GPC3, (<median/≥median), n	14/8	0.402
ALB, (<median/≥median), n	11/11	0.272
**CTC counts (FACS)**		
CTC counts, /5 mL, (<150/≥150), n	11/11	0.492
**Tumor biopsy**		
Tumor differentiation, (well/moderate, poor), n	5/6	0.353
PD-L1, H-score, (<10/≥10), n	6/5	0.238
**Clinical features**		
Age, y, (<75/≥75), n	12/10	0.852
Gender, (male/female), n	17/5	0.321
ECOG PS, (0/≥1), n	17/5	0.945
Etiology, (HBV, HCV/NBNC), n	12/10	0.212
mALBI grade, (1,2a/2b,3), n	11/11	0.767
AFP, ng/mL, (<13.4/≥13.4), n	9/13	0.974
DCP, mAU/mL, (<100/≥100), n	8/14	0.099
Prior systemic therapy, (TKI/none or HAIC), n	8/14	0.525
Extrahepatic metastasis, (present/absent), n	5/17	0.442
Vascular invasion, (present/absent), n	5/17	0.197
Maximum tumor size, cm, (<4/≥4), n	13/9	0.468
Tumor number, (<3/≥3), n	4/18	0.738
BCLC stage (B/C), n	11/11	0.600

Abbreviations: AFP, a-fetoprotein; BCLC stage, Barcelona Clinic Liver Cancer stage; CTC, circulating tumor cell; DCP, des-gamma-carboxy prothrombin; ECOG PS, Eastern Cooperative Oncology Group performance status; HAIC, hepatic arterial infusion chemotherapy; H-score, histological score; IHC, immunohistochemistry; mALBI, modified albumin-bilirubin; NBNC, nonB-nonC; PD-L1, programmed death ligand 1; qRT-PCR, quantitative reverse-transcription polymerase chain reaction; TKI, tyrosine kinase inhibitor. Bold: not significant.

## Data Availability

The data of the current study are available from the corresponding author upon reasonable request.

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
