# Peer review of "Programmed Death Ligand 1 Expression in Circulating Tumor Cells as a Predictor and Monitor of Response to Atezolizumab plus Bevacizumab Treatment in Patients with Hepatocellular Carcinoma"

_cancers, 2024, doi:10.3390/cancers16091785_

Round 1

Reviewer 1 Report

Comments and Suggestions for Authors

There are a few comments:

Despite the small sample size, the methods are comprehensively described, and the interpretations seem appropriate.

It would be beneficial to specify whether the tumor biopsies were needle biopsies or another type.

Consider enhancing the manuscript by including PD-L1 expression patterns in tumor biopsy specimens and correlating these patterns with the drug response.

It would be better to include radiological imaging results to evaluate treatment response.

Comments on the Quality of English Language

Please check for any English grammar and spelling errors.

For example, Table 2. Antiobodies ->Table 2. Antibodies

Author Response

It would be beneficial to specify whether the tumor biopsies were needle biopsies or another type.

We appreciate your suggestion. HCC tissue specimens were obtained from the center of the tumor using a percutaneous fine-needle aspiration biopsy under abdominal ultrasound guidance. Accordingly, we added the description to the Results section (page 10, lines 297-299).

Consider enhancing the manuscript by including PD-L1 expression patterns in tumor biopsy specimens and correlating these patterns with the drug response.

We appreciate your important suggestion. The correlation between PD-L1 H score of liver tumor biopsies and Atezo/Bev therapeutic response was evaluated. “There was no difference in PD-L1 H-score of liver tumor biopsies by Atezo/Bev treatment effect (Figure 3C). These results indicate that the intensity of PD-L1 expression in biopsy specimens of primary liver tumors does not correlate with PD-L1 expression in CTCs and is poorly associated with the therapeutic response to Atezo/Bev.” Accordingly, we added the description to the Results section (page 11, lines 302-306) and the figure and legends to Figure 3C.

It would be better to include radiological imaging results to evaluate treatment response.

I appreciate your comments. Radiological response was evaluated by dynamic CT or Gd-EOB-MRI at 8-12 weeks after the first administration using RECIST ver. 1.1 to determine PR/SD and PD (page 4, lines 118-122). Changes of tumor size (%) in 22 HCC patients treated with Atezo/Bev were added to Figure 1D. Accordingly, we added the figure and legends to Figure 1D.

Please check for any English grammar and spelling errors.

For example, Table 2. Antiobodies ->Table 2. Antibodies

I appreciate your pointing out the spelling errors. Typos were corrected (page 5, lines 157).

Reviewer 2 Report

Comments and Suggestions for Authors

In this report the authors measure PD-L1 and other markers expression in an attempt to see if they could be prognosticators of treatment. The authors use a battery of analysis and statistical tests to test this hypothesis. There is ample data here and the paper has value. However, as it is presented here it is somewhat hard to understand and follow their experimental setup and conclusions. Additionally, in the interest of transparency, the authors should provide supplemental tables with detailed information on how they ran the statistical tests. It is hard to evaluate the claims without access to complete data. Following are more specific comments.

Major issues

1.       The authors should provide a supplemental table with all statistical calculations. Section 2.12 in the introduction mentions the test used only broadly but this reviewer tried to replicate some statistical analysis but found it hard to do. For example, for the data in Table 4, I assume that the Univariate Cox proportional test was used but what exact values were used to extract the p-values? What does the column named “patients” represent? The authors assert that PD-L1 expression was associated with PFS in the results text but the column lists 11/11 for multiple factors not just PD-L1. Why is there a difference in p-values?

2.       What test was used in Figure 2A to show statistical significance for PD-L1 but not the other genes?

3.       Section 3.3 in the results is confusing. The authors should expand the results text in this section. What is the point of this study and if there is no association, what is the conclusion? Since this section follows the RNA expression section, are the authors saying that although there is differential RNA expression, there is no effect on translation? As it is written, it is hard to understand the point of this result.

4.       Section 3.4 goes back to RNA expression. If the previous section concluded that RNA expression does not translate (no pun intended!) to protein expression, what then is the significance of the RNA expression?

5.       Lines 382-386 discuss how PD-L1 expressing cells may have more aggressive characteristics, etc.. Are these studies looking at protein expression or RNA? If there are no changes in protein expression, how is the immune system detecting the differential expression? Is the immune system surveilling RNA levels in the cells?

6.       The general conclusion of the paper is confusing. Are the authors concluding that changes in PD-L1 RNA levels can be used to classify these tumors? Are they suggesting that some patient tumors can be diagnosed/classified by RNA expression but others not? Are they suggesting that RNA expression changes can be used as a prognosticator for treatment response? Take for example this statement: “Analysis of CTC-derived RNA collected serially during Atezo/Bev treatment in patients with HCC suggested that patients with higher PD-L1 expression in CTCs at baseline were more responsive to treatment.” When the authors say “suggested” do they mean that they found this in this study or that it was suggested in the literature? And “more responsive to treatment” is somewhat subjective. A stronger statement should be made based on statistical analysis (e.g. twice as responsive, 3 fold more responsive, etc).  

Minor issues

1.       Please use the entire name for CTC in the introduction at least once before using the acronym. The reader should not have to go back to the abstract to figure out what it is.

Section 3.1 first paragraph. Please indicate with numbers in the text how many patients were analyzed prior to treatment. All 22 patients in table 1, some of them, etc? This information is lost through in results.

Author Response

Major issues

  1. The authors should provide a supplemental table with all statistical calculations. Section 2.12 in the introduction mentions the test used only broadly but this reviewer tried to replicate some statistical analysis but found it hard to do. For example, for the data in Table 4, I assume that the Univariate Cox proportional test was used but what exact values were used to extract the p-values? What does the column named “patients” represent? The authors assert that PD-L1 expression was associated with PFS in the results text but the column lists 11/11 for multiple factors not just PD-L1. Why is there a difference in p-values?

We appreciate your valuable suggestions. Patient data regarding Table 4 is attached. This data is represented in a heat map in Figure 1D. Based on the grouping of each factor in Figure 1D (e.g., age <75/ ≥75), the factors in Table 4 are divided into two groups. The attached excel data contains and links to detailed patient privacy data including age, gender, etiology, and survival time. Therefore, we refrained from disclosing information. However, we consider the clinical background and CTC characteristics of each case to be represented in Figure 1D. In Figure 1D and Table 4, 22 cases were divided into two groups at the median in the CTC RNA expression (qRT-PCR) category. Therefore, many factors are divided into 11/11 cases. “Patients” was changed to “Patients number” and “n” was added to each factor in Table 4. Univariate log-rank test was used for the tests in Table 4, so the text was modified (page 6, lines 225-226).

  1. What test was used in Figure 2A to show statistical significance for PD-L1 but not the other genes?

We appreciate your pointing. In Figure 2A, the Mann-Whitney test was used to show the statistical significance of RNA expression of 14 genes in CTCs before treatment with Atezo/Bev in the PR/SD and PD groups. The Mann-Whitney test was used, which is described in the legend of Figure 2.

  1. Section 3.3 in the results is confusing. The authors should expand the results text in this section. What is the point of this study and if there is no association, what is the conclusion? Since this section follows the RNA expression section, are the authors saying that although there is differential RNA expression, there is no effect on translation? As it is written, it is hard to understand the point of this result.

We appreciate your important suggestions. We apologize for the inadequate explanation of the difference in characteristics between primary liver tissue and CTCs, and have added the following text. “Cancer cells can undergo epithelial-to-mesenchymal transition (EMT) to facilitate their detachment from the primary tumor and intravasation into the blood circulation [12,13]. The characteristics of CTCs differ from those of tumor cells in primary tumors because EMT involves the loss of epithelial characteristics, for example, downregulation of the adhesion molecule E-cadherin, and the acquisition of mesenchymal characteristics, including expression of the cytoskeletal protein vimentin [12,13].” Accordingly, we added the description to the Introduction section (page 2, lines 64-70).

In the study of PD-L1 expression in primary tumors and CTCs, “as PD-L1 expression varies between primary and metastatic sites and even within tumor nests, PD-L1 expression analysis of CTCs is considered a useful tool to complement this heterogeneity [34]” (page 15, lines 410-412).

In response to Reviewer 1's suggestion, we evaluated the correlation between the PD-L1 H score of liver tumor biopsies and the therapeutic effect of Atezo/Bev. “There was no difference in PD-L1 H-score of liver tumor biopsies by Atezo/Bev treatment effect (Figure 3C).” The following description was added. ”These results indicate that the intensity of PD-L1 expression in biopsy specimens of primary liver tumors does not correlate with PD-L1 expression in CTCs and is poorly associated with the therapeutic response to Atezo/Bev.” Accordingly, we added the description to the Results section (page 11, lines 302-306) and the figure and legends to Figure 3C.

  1. Section 3.4 goes back to RNA expression. If the previous section concluded that RNA expression does not translate (no pun intended!) to protein expression, what then is the significance of the RNA expression?

We appreciate your suggestions. As responded to above comments, the characteristics of CTCs differ from those of tumor cells in primary tumors.

In this study, analysis of CTC-derived RNA collected serially during Atezo/Bev treatment in patients with HCC indicated that patients with higher PD-L1 expression in CTCs at baseline were more responsive to treatment. In addition, PD-L1 expression in CTCs is dynamically altered by Atezo/Bev treatment, decreasing the effective response and increasing progression. Furthermore, PD-L1 expression in primary liver tumors did not correlate with PD-L1 expression in CTCs and was not associated with therapeutic response to Atezo/Bev.

  1. Lines 382-386 discuss how PD-L1 expressing cells may have more aggressive characteristics, etc.. Are these studies looking at protein expression or RNA? If there are no changes in protein expression, how is the immune system detecting the differential expression? Is the immune system surveilling RNA levels in the cells?

We appreciate your pointing. These studies examine protein expression. The immune system detects protein expression, not at the RNA level.

In this study, we analyzed 14 genes related to PD-L1, stem cells, and EMT to evaluate gene expression in CTCs related to the therapeutic effect of Atezo/Bev and found PD-L1 is a predictor of the treatment. In CTCs, which are rare tumor cells, it is reasonable to analyze RNA in order to objectively analyze 14 genes at the same time. In this study, we evaluated PD-L1 protein expression by FACS only in a few cases and found that it was consistent with changes in RNA levels (Figure 5A). We then cited these references to discuss the significance of the increased PD-L1 expression of CTCs in Atezo/Bev-resistant cases.

  1. The general conclusion of the paper is confusing. Are the authors concluding that changes in PD-L1 RNA levels can be used to classify these tumors? Are they suggesting that some patient tumors can be diagnosed/classified by RNA expression but others not? Are they suggesting that RNA expression changes can be used as a prognosticator for treatment response? Take for example this statement: “Analysis of CTC-derived RNA collected serially during Atezo/Bev treatment in patients with HCC suggested that patients with higher PD-L1 expression in CTCs at baseline were more responsive to treatment.” When the authors say “suggested” do they mean that they found this in this study or that it was suggested in the literature? And “more responsive to treatment” is somewhat subjective. A stronger statement should be made based on statistical analysis (e.g. twice as responsive, 3 fold more responsive, etc).

We appreciate your valuable suggestions. In Section 3.2., we analyzed the Median PFS of the CTC PD-L1 High and Low groups at baseline and added the description “Median PFS, CTC PD-L1 High/Low 11.97/3.09 months” (page 10, lines 280-281). Based on the results, the conclusions were revised as follows. “Analysis of CTC-derived RNA collected serially during Atezo/Bev treatment in patients with HCC indicated that patients with higher PD-L1 expression in CTCs at baseline were 3.9 folds more responsive to treatment” (page 16, lines 422-424 and page 1, lines 36-38).

Minor issues

  1. Please use the entire name for CTC in the introduction at least once before using the acronym. The reader should not have to go back to the abstract to figure out what it is.

I appreciate your pointing. In the introduction, the entire name of the CTC was added (page 2, lines 62).

Section 3.1 first paragraph. Please indicate with numbers in the text how many patients were analyzed prior to treatment. All 22 patients in table 1, some of them, etc? This information is lost through in results.

I appreciate your remarks. Analysis of 1), 2), and 4) was performed on all 22 patients. For 3), liver tumor biopsy tissue was obtained and analyzed before treatment in 11 of the 22 patients. Accordingly, we added the description to the Results section (page 7, lines 243-245).

Round 2

Reviewer 2 Report

Comments and Suggestions for Authors

The authors made the changes I requested. This reviewer is satisfied.